# Studying Genetic Diversity and Relationships between Mountainous Meihua Chickens Using Mitochondrial DNA Control Region

**DOI:** 10.3390/genes14050998

**Published:** 2023-04-28

**Authors:** Bo Ran, Wei Zhu, Xiaoling Zhao, Linxiang Li, Zhixin Yi, Miao Li, Tao Wang, Diyan Li

**Affiliations:** 1School of Pharmacy, Chengdu University, Chengdu 610106, China; ranbo1998@163.com (B.R.);; 2College of Animal Science and Technology, Sichuan Agricultural University, Chengdu 611130, China; weizhu451@163.com (W.Z.);; 3Bazhong Academy of Agricultural and Forestry Sciences, Bazhong 610066, China

**Keywords:** Mountainous Meihua chicken, mtDNA, D-loop, genetic diversity, maternal lineage

## Abstract

The Mountainous Meihua chicken is a unique regional germplasm resource from Tongjiang County, Bazhong City, China, but its genetic structure and evolutionary relationships with other native chicken breeds in the Sichuan region remain unclear. Here, we analyzed a total of 469 sequences, including 199 Mountainous Meihua chicken sequences generated in this study, together with 30 sequences representing 13 clades and 240 sequences from seven different Sichuan local chicken breeds downloaded from NCBI. These sequences were further used to analyze genetic diversity, patterns of population differentiation, and phylogenetic relationships between groups. We show that Mountainous Meihua chicken mtDNA sequences have high haplotypic and nucleotide diversity (0.876 and 0.012, respectively) and with a T bias that is suggestive of good breeding potential. Phylogenetic analysis showed that Mountainous Meihua chickens belong to clades A, B, E, and G and have a low affinity to other chicken breeds, with a moderate degree of differentiation. A non-significant Tajima’s D indicates that no demographic expansions occurred in the past. Finally, the four maternal lineages identified in Mountainous Meihua chicken showed unique genetic characteristics.

## 1. Introduction

The Bazhong Mountainous Meihua chicken (*Galus galus domesticus*) is a local genetic resource with ~2000 years of history in the Qinba mountain region [1]. In 2013, a general survey of germplasm resources showed that Mountainous Meihua chickens are primarily distributed in Nanjiang County, Bazhong City, in particular across 10 townships (e.g., Shaba, Huitan, and Hongyan of Nanjiang County) [2]. The unique shape, moderate growth speed, and excellent egg and meat quality make Mountainous Meihua chickens a highly valuable resource [3]. The breed is also resistant to rough feeding and possesses a good foraging ability, adaptability, and disease resistance [4], all of which are important traits for improving local breeds. In recent years, and thanks to the support of local municipal governments, seed conservation farms meant for planned seed selection and breeding have been established [5]. Now, there is one protected population, 68 protected families, and more than 2000 pure chickens of this breed.

As it is an important extracellular genetic information carrier, mitochondrial DNA (mtDNA) has low relative molecular weights, high copy numbers, maternal inheritance, and no/rare recombination events, which means that it is often used as a marker in many molecular studies [6,7,8,9]. It is a popular choice for phylogenetic, population genetic, and phylogeographic analyses [10]. The mtDNA D-loop region is a non-coding region rich in A/T bases, which can be recognized by specific molecules such as RNA polymerase and used as the starting sequence of transcription [11]. In the mitochondrial genome, coding genes are mainly used for the biodiversity research of different species, while non-coding D-loop regions are commonly used for genetic diversity analysis and maternal origin research due to their evolution speed being 5–10 times higher than that of other regions on mtDNA [12,13,14].

Using mtDNA information, previous studies indicated that domestic chickens independently originated from a group of red jungle poultry found in forests in Southeast Asia and India and gradually distributed throughout the world with continuous domestication and transmission, resulting in many new breeds [15]. Research on the origin of Chinese domestic chicken breeds revealed that the red jungle fowl from Thailand and its neighboring areas are considered to be the maternal ancestors of Chinese chicken breeds [16]. However, a series of independent domestication events occurred across Asia and the Indian subcontinent, supporting the theory of there being multiple origins of domestic chickens [17,18]. In 2006, Liu et al. [19] first proposed that domestic chickens in Eurasia are distributed among nine highly different mtDNA evolutionary branches (A–I) and pointed out different regions where different branches may originate. Later, other researchers proposed more branches and further determined the maternal origin of domestic chickens [20]. Analysis of the origin of Japanese indigenous chickens revealed that the wild red jungle fowl was domesticated in Southeast Asia and passed into China to form non-wild clades (A and B), which together form the foundation of Japanese indigenous chickens [21]. The research on the origins and transmission routes of domestic chickens in the Middle East and Africa supports the hypothesis that the Indian subcontinent is the maternal origin of these chickens. Additionally, ancient and modern trade networks contributed to the modern diversity of native chickens [22]. A growing number of studies on mtDNA D-loop regions have revealed genetic diversity within populations and inter-population relationships of some new local chicken breeds in various regions in recent years [23,24,25,26], which have provided important support for modern breeding or chicken genetic diversity protection plans.

Mountainous Meihua chickens have been genetically characterized in the past, in particular for phenotypes associated with growth and development and muscle and egg quality [3]. However, the evolutionary mechanisms of differentiation for this chicken breed have yet not been fully explored. Molecular technology can efficiently and accurately study the genetic differences among breeds of animals. Mitochondrial DNA (mtDNA) polymorphisms in various animal populations can help to clarify ancestral lineages or compare different populations, which has also been shown in chickens [27]. Here, we analyzed the partial mtDNA control region sequences of Mountainous Meihua chickens and seven other chicken breeds sampled in the Sichuan Province of China, with the goal to assess the levels of genetic diversity and population structure to uncover ancestral maternal lineages. Our results provide insights into the evolutionary history of Meihua chickens and other local breeds, which can help to improve this valuable genetic resource.

## 2. Materials and Methods

### 2.1. Ethical Statement

The collection of blood samples from Mountainous Meihua chickens was performed according to the Guidelines for Experimental Animals established by the Ministry of Science and Technology (Beijing, China, revised in March 2017). Ethical approval was provided by the Ethics Committee of the Sichuan Agricultural University (protocol number 2020202010).

### 2.2. Sample Collection and DNA Extraction

Blood samples were collected from 199 Mountainous Meihua chickens via wing veins. These chickens were all adult males in the Mountainous Meihua chickens breeding base in the Shuangquan Township, Tongjiang County. The blood samples were stored at −20 degrees to ensure the integrity of genomic DNA. These samples were then sent to the College of Animal Science and Technology at Chengdu Campus of Sichuan Agricultural University for genomic DNA and further genetic analysis. A proteinase-K/Phenol-chloroform standard procedure was used to extract genomic DNA, as previously described [28]. Isolated DNA samples were measured in terms of concentration and assessed in terms of quality using a NanoDrop 2000 ultramicroscopic spectrophotometer.

### 2.3. PCR Amplification and DNA Sequencing

The following forward (5′-AGGACTACGGCTTGAAAAGC-3′) [29] and reverse (5′-ATGTGCCTGACCGAGGAACCAG-3′) [19] primers were used for amplification of the mtDNA control region (approximately 500 bp) of Mountainous Meihua chicken. PCR amplification was performed using a 25 μL solution containing 12.5 μL of 2 × Taq PCR Master Mix, 1.25 μL of template genomic DNA (200 ng), 1.25 μL of each primer (10 pmol/μL), and 8.75 μL of ddH_2_O. After incubation at 95 °C for 3 min, 33 cycles were performed, as follows: 94 °C for 35 s, 58.4 °C for 35 s, and 72 °C for 1 min, with a final extension at 72 °C for 10 min. PCR products were purified using the gel recovery kit and sequenced using an ABI 3730 automated sequencer (Applied Biosystems, Carlsbad, CA, USA).

### 2.4. Data Analysis

The Nucleotide Basic Local Alignment Search Tool (BLASTN) was used to ensure the accuracy of the sequenced DNA fragments. The mitochondrial D-loop sequences of seven other Sichuan chicken breeds [30] were downloaded from NCBI (GenBank accession numbers: N381367 and MN381709), including 27 sequences for Pengxian yellow chickens (PX), 29 sequences for Jinyang silky chickens (JS), 30 sequences for Emei Black chickens (EM), 40 sequences for Jiuyuan black chickens (JY), 29 sequences for Muchuan silky chickens (MC), 39 sequences for Miyi chickens (MY), and 46 sequences for Shimian Caoke chickens (SM). The geographical distribution of the 8 chicken breeds is shown in Figure 1. The obtained 439 sequences were aligned using ClustalX 2.1 in the DNASTAR software package and truncated to the longest common sequence (482 bp) among all samples for further analysis. Taking the red jungle fowl (NC_007235) as the reference sequence, we analyzed genetic variation using DnaSP vs6.0 and defined haplotypes, extracted variation site information (V), and calculated the haplotype diversity (Hd), nucleotide diversity (π), the average number of nucleotide differences (K), Tajima’s D values, Nucleotide divergence (Dxy), net genetic distance (Da), Genetic differentiation index (Gst) and gene flow (Nst), haplotypes, and average no. of nucleotide differences. To evaluate sequence variation among and within populations, we conducted analysis of molecular variance (AMOVA) and pairwise fixation index (Fst) using Arlequin version 3.5 [31]. MEGA 6.0 [32] was used to analyze sequence base composition and establish phylogenetic relationships among the 8 chicken breeds. A total of 30 mitochondrial D-loop sequences downloaded from NCBI, representing a total of 13 clades, were used to analyze the maternal origin of the populations, following previous studies [20]. An unrooted neighbor-joining (NJ) tree was constructed using the Kimura 2-parameter model. Bootstrap values of the phylogenetic tree were estimated from 1000 replicates. A maximum parsimony median-joining (MJ) network was made of the 439 sequences using the software PopART [33].

## 3. Results

### 3.1. Sequence Variation and Population Diversity

The mtDNA of 439 individuals from eight chicken breeds was truncated to 482 bp. The observed base compositions of the mtDNA D-Loop in MH were 30.51% (T), 28.09% (C), 28.38% (A), and 13.02% (G) (Table 1), showing an obvious A/T bias. Other chicken breeds showed similar base composition signatures.

We identified 46 polymorphic sites, of which 39 were parsimony informative ones, and seven were singleton variable sites. These polymorphisms defined a total of 44 haplotypes (Figure 2), ranging from 5 (MC) to 23 (MH) in each breed. We also found 17 haplotypes shared among multiple populations (Table 2), in particular MY, SM, and MC. The haplotypes found in MH mainly include H2, H3, H5, H7, H10, H12, and H20, among which H3, H5, H7, H12, and H20 were shared haplotypes, accounting for 53.77% of the total number of MH samples analyzed. In contrast, H2 and H10 were unique MH haplotypes and accounted for 31.16% of the total samples.

Subsequently, we analyzed the overall genetic diversity indices in MH (Table 3) and contrasted them with the corresponding information available for other chicken breeds [30]. A total of 28 polymorphic sites were present in MH, of which 25 were parsimony informative sites and 3 were singletons. MH haplotype (0.876) and nucleotide (0.012) diversity, as well as the average number of nucleotide differences (5.936), were in the upper range of all chicken breeds studied. The estimated Tajima’s D across all breeds was not significant, indicating that no significant demographic expansions occurred in the past.

The genetic variation within and between population was quantified via AMOVA based on Kimura-2 parameter distances (Table 4) and showed that most genetic diversity was found within breeds (81.50%). Notwithstanding, we found that the differences between populations were highly significant (*p* < 0.01).

### 3.2. Nucleotide Divergence and Genetic Differentiation between Populations

We analyzed the average number of nucleotide substitutions per site (Dxy) and the net divergence (Da) between populations based on the mtDNA control region (Table 5). We estimated that the Dxy was between the eight chicken breeds ranged between 0.930 (PX) and 1.710 (JS), while the maximum Da was between MC and JY (0.735), and the minimum degree of differentiation was between JS and PX (−0.027). The Dxy between MH and other chicken breeds ranged from 1.212 to 1.652, with the Da ranging from 0.163 to 0.578.

We also estimated the coefficient of genetic differentiation (Gst) and gene flow (Nst) between eight populations (Table 5) and found the maximum degree of differentiation was between SM and PX and minimum degree of differentiation was between JS and PX. Furthermore, MH was the greatest genetic distance away from MC (Nst = 37.888) and had the most affinity with JS (Nst = 11.728).

### 3.3. Phylogenetic Relationships and Haplotype Network Using the mtDNA D-Loop

Phylogenetic analysis was conducted to estimate evolutionary genetic distances using the maximum composite likelihood method available in the MEGA6.0 software. Phylogenetic analysis (Figure 3) revealed 44 haplotypes in indigenous chickens that were divided across 5 (A, B, C, E, and G) of the 13 clades (clades A–I and W–Z), following the classification of Liu [19] and Miao [20]. The five clades, A, B, C, E, and G, contained 21, 6, 3, 9, and 5 haplotypes, respectively. MH chicken haplotypes were distributed across the A, B, E, and G clades. We further analyzed the haplotype composition in MH chicken breeds (Table 6), which showed that 29.65% of the samples belong to the A clade, 41.71% belong to the B clade, 27.14% belong to the E clade, and 1.50% belong to the G clade.

A maximum parsimony median-joining network was constructed based on the 44 haplotypes to further determine the phylogenetic relationship and genetic distances among the eight native chicken breeds. Network analysis depicts the minimum mutational distances among the haplotypes, and these are consistent with features observed in the neighbor-joining tree (Figure 4). By contrast, the haplotypes present within clades are closer to each other than those identified between clades are. Clades A and E are shared by all chicken breeds.

## 4. Discussion

Here, we presented an examination of the genetic diversity and maternal origin of Mountainous Meihua chickens (MH) and explored the genetic differentiation and evolutionary relationships between MH and seven other Sichuan chicken breeds. The analysis of the base composition of the mitochondrial D-loop indicated an A/T bias across all eight chicken breeds, which is consistent with studies in some other chicken breeds [25,34]. Most previous studies found that variable sites in the D-loop region were in 133–446 bp, as the mutation rate in this region is higher than it is in the other regions [35]. In our study, most of the genetic variation among the eight breeds was also located in this region. However, some variation sites found by Yin [36] were located at positions 711, 1214, 1215, and 1222 in the Lueyang Black-bone Chicken, and at positions 686, 792, and 1215 in the Jiangxi Anyi Tile-like Chicken [37]. As a result, longer sequences are likely to provide more comprehensive information on variation sites than shorter sequences will.

Studies have shown that commercial breeds have relatively low genetic diversity, as only a few sires produce many offspring [38]. Native chicken breeds, however, become good genetic resources for future breeding needs in order to create better varieties for commercial use due to their high level of genetic variation [39,40]. The ability of species to adapt to harsh environments and resist catching diseases can be improved by the genetic diversity within them [35]. Haplotype and nucleotide diversities are two key indicators to evaluate genetic variation, with higher values being indicative of higher genetic diversity [41]. In MH, we found haplotype and nucleotide diversities of 0.876 and 0.012, respectively, which are higher than those of most of local breeds here analyzed; it is important to note that the sample sizes of different breeds in our study are inconsistent. In order to properly compare the samples with different sizes, those conducting future studies should use the rarefaction method. Similar results were previously found when comparing MH with other indigenous chicken breeds from China using the same metrics [42,43]. These results could provide further support for improving breeding practices.

Tajima’s D values were not significant across all chicken breeds, indicating that neither balancing nor purifying selection occurred in these chicken breeds [30], and thus, they evolved neutrally. This is consistent with observations of native flocks in Thailand and the Philippines [17,44]. MH were raised as free-range backyard flocks for a long time, with episodes of random mating, and little to no control of breeding practices. These factors may have reduced artificial selective pressures, and thus, support neutral evolution. Further hierarchical analysis of molecular variance, AMOVA, indicated that the differentiation among maternal lineages in the eight chicken populations mainly resulted from variation within populations, which is similar to previous findings by Wani [7] and Teinlek [44].

Gst and Nst are commonly used to characterize genetic differentiation and gene exchanges between populations. Here, we found the highest degree of differentiation between MH and JY, and the lowest one between MH and MY. Similarly, while gene exchanges between MH and MC were frequent, those between MH and JS were rare. Our results showed large differences between MH and MY, and little diversification between MH and JY. In addition, we observed the highest Da value between MH and MC, and the lowest one between MH and EM. These differences may result from different living conditions, selection pressures, geographical isolation, and breeding histories [45,46,47,48,49,50].

According to Liu [19] and Miao [20], the mtDNA phylogenetic tree of domestic chicken and the red jungle fowl can be divided into 13 clades. Of these, clades A and B are widely distributed across the world and native to Yunnan and the surrounding areas. Clade C is endemic to the Huang-Huai River Basin in China, clade G is distributed in Yunnan and the adjacent areas, and clade E is common among Indian and European domestic chickens. A study on the matrilineal origin of Indonesian chickens showed that Indonesian chickens were distributed among haplogroups B, D, and E [51]. The phylogenetic results of a study of Bangladesh native chickens showed five distinct mitochondrial haplogroups (A, D, E, F, and I), suggesting that they have a more maternal origin [10]. A previous study indicated that the mtDNA D-loop sequences of seven Sichuan native chickens were clustered in clade B [30], but we found these groups include five clades, specifically A, B, C, E, and G. Clades A, B, and E were the dominant lineages in the MH group, with a frequency of 98.5%, with clade E representing 1.5%, which could have resulted from admixture with local Yunnan varieties. Interestingly, this clade is represented by two haplotypes. One is the most frequent clade, G haplotype, and the other one (detected only in MH) diverges from it, and it can be identified on the basis of one substitution. This could be indicative that this admixture happened a long time ago since the new haplotype detected only in MH had time to evolve. Unlike other chicken populations, a considerable part of the MH population (41.71%) belong to haplogroup B, showing characteristics that are different from those of other chicken breeds. In addition, maternal lineage sharing has been reported among different indigenous chicken breeds in various geographical locations [52,53,54]. Among the eight flocks studied, clades A and E were found to be in common, indicating a common origin.

In summary, our study suggests that Mountainous Meihua chickens have a relatively high level of genetic diversity and present a certain degree of genetic differentiation from seven other native breeds from China. The results also showed that there were four maternal lineages in this breed, specifically belonging to clades A, B, and E. Further genomic analyses should be conducted in the future to provide further understanding of this unique breed.

## Figures and Tables

**Figure 1 genes-14-00998-f001:**
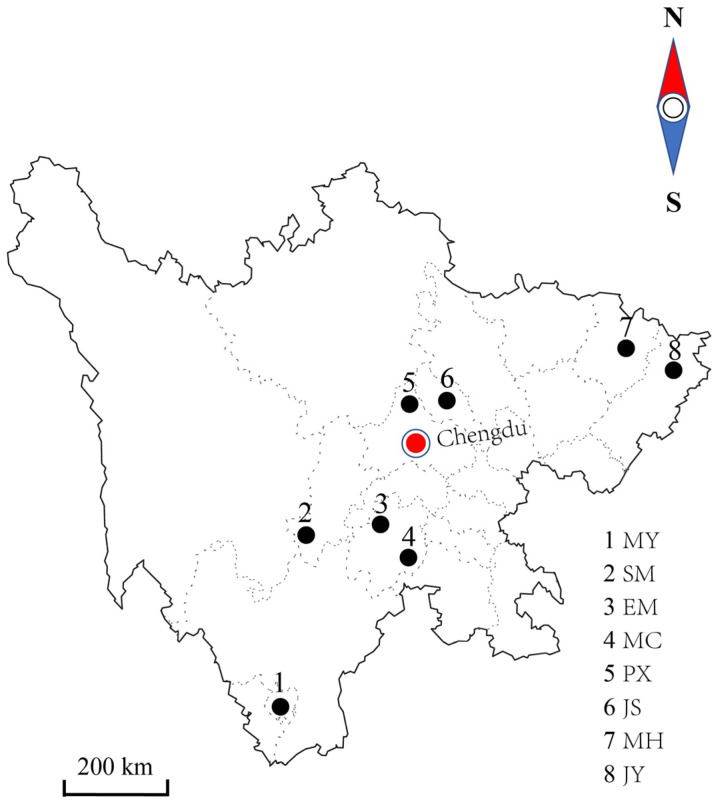
Distribution map of eight local chicken breeds in Sichuan. Note: MY: Miyi chickens; SM: Shimian Caoke chicken; EM: Emei Black chicken; MC: Muchuan silky chicken; PX: Pengxian yellow chicken; JS: Jinyang silky chicken; MH: Mountainous Meihua chicken; JY: Jiuyuan black chicken. The red dot Chengdu is capital of Sichuan province.

**Figure 2 genes-14-00998-f002:**
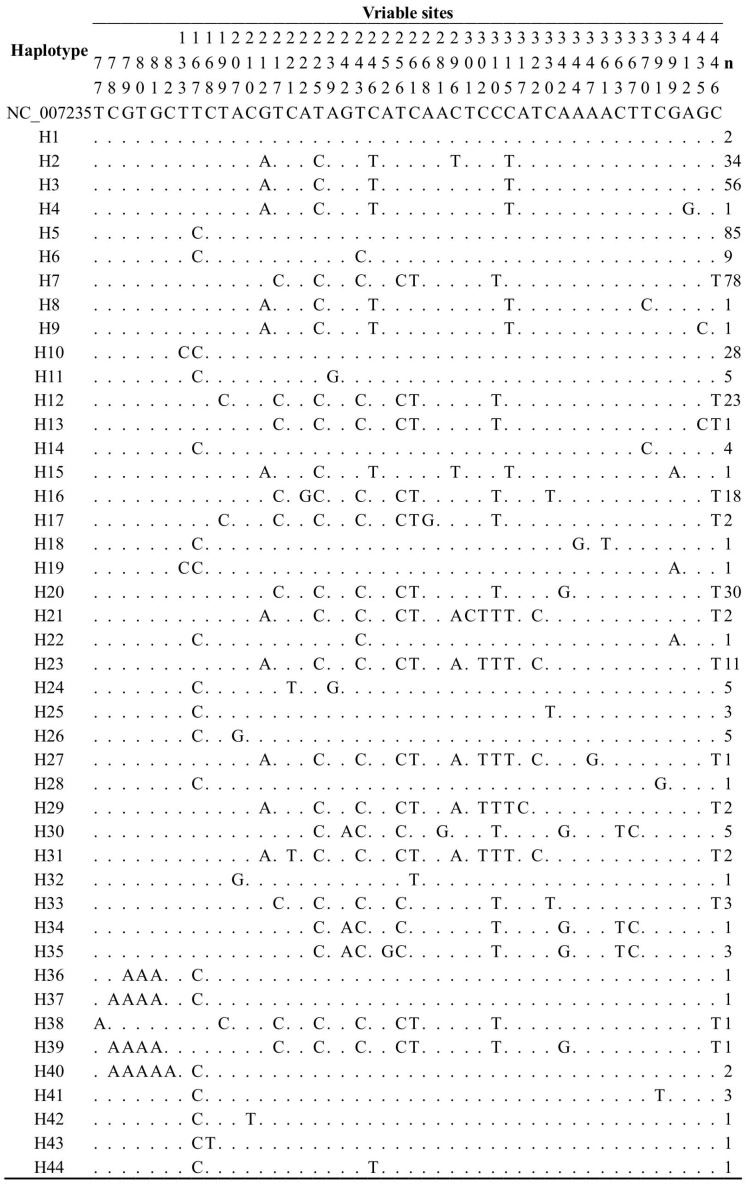
Variable information of the 8 Sichuan chicken breeds. Note: These numbers indicated mutation position. “n” represents the number of individuals per haplotype. Dot (.) represents the identical nucleotide with the reference sequence (NC_007235).

**Figure 3 genes-14-00998-f003:**
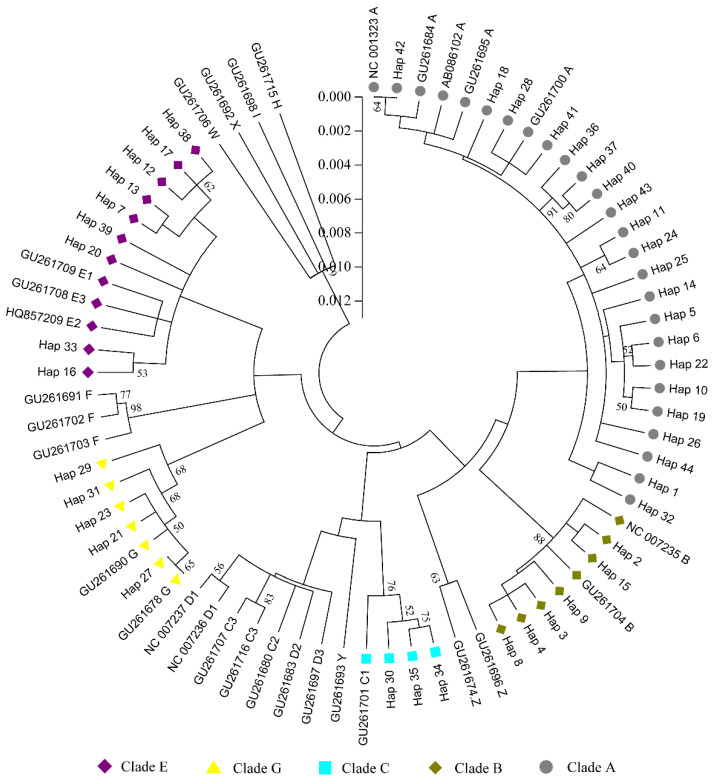
Phylogenetic tree of mitochondrial D-loop nucleotide sequences based on 74 haplotypes (30 from reference DNA Genbank representing 13 clades [20] and 44 from indigenous chicken breeds). The evolutionary history was inferred using the neighbor-joining method. The numeral at each branch indicates the bootstrap value of replications (1000 replicates). Bootstrap values lower than 50% are not shown.

**Figure 4 genes-14-00998-f004:**
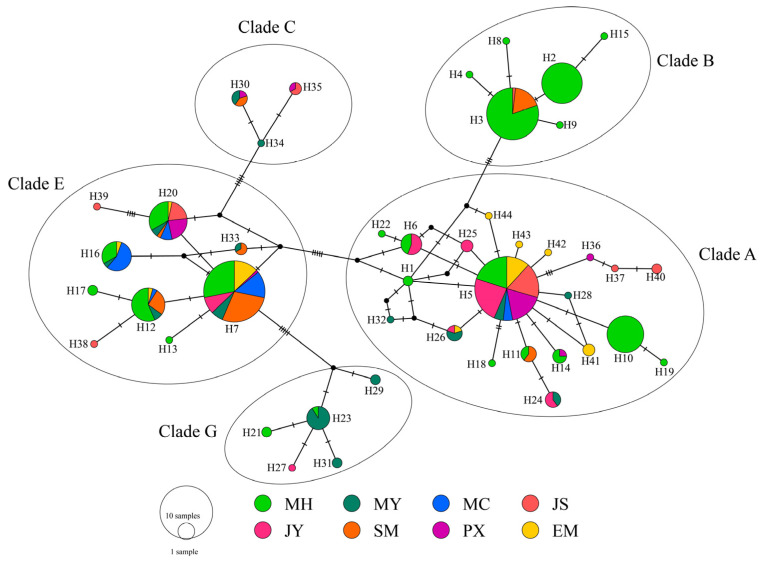
Maximum parsimony median-joining network of MH with other Sichuan local chicken breeds based on 482 bp control-region sequence. Node sizes are proportional to haplotype frequencies. The lines linking the nodes are proportional to the mutation steps. Different colors within each haplotype ring correspond to different chicken breeds. Five clades (A, B, C, E and G) were formed.

**Table 1 genes-14-00998-t001:** The base composition of mtDNA D-Loop sequence.

Breed	Number of Individuals	Total Number of Bases	Content %					
			T(U)	C	A	G	A + T	G + C
MH	199	482	30.51	28.09	28.38	13.02	58.89	41.11
JY	40	482	30.43	28.10	28.32	13.15	58.75	41.25
MY	39	482	30.59	27.88	28.41	13.11	59.00	41.00
SM	46	482	30.47	28.07	28.35	13.11	58.82	41.18
MC	29	482	30.48	28.06	28.24	13.22	58.72	41.28
PX	27	482	30.38	28.15	28.29	13.18	58.67	41.33
JS	29	482	30.37	28.09	28.41	13.13	58.79	41.21
EM	30	482	30.46	28.08	28.31	13.15	58.77	41.23

**Table 2 genes-14-00998-t002:** Distribution of 44 haplotypes in 8 Sichuan chicken breeds.

MH(199)	JY(40)	MY(39)	SM(46)	MC(29)	PX(27)	JS(29)	EM(30)
H1(2)	H5(20)	H5(4)	H3(10)	H5(4)	H5(15)	H3(1)	H5(10)
H2(34)	H6(5)	H7(5)	H7(22)	H7(11)	H7(1)	H5(15)	H7(10)
H3(45)	H7(7)	H12(2)	H11(3)	H12(1)	H14(1)	H20(6)	H12(1)
H4(1)	H24(3)	H16(1)	H12(6)	H16(10)	H20(7)	H35(2)	H16(1)
H5(17)	H25(3)	H20(2)	H20(1)	H20(3)	H30(1)	H37(1)	H20(1)
H6(4)	H26(1)	H23(10)	H30(2)		H35(1)	H38(1)	H26(1)
H7(22)	H27(1)	H24(2)	H33(2)		H36(1)	H39(1)	H41(3)
H8(1)		H26(3)				H40(2)	H42(1)
H9(1)		H28(1)					H43(1)
H10(28)		H29(2)					H44(1)
H11(2)		H30(2)					
H12(13)		H31(2)					
H13(1)		H32(1)					
H14(3)		H33(1)					
H15(1)		H34(1)					
H16(6)							
H17(2)							
H18(1)							
H19(1)							
H20(10)							
H21(2)							
H22(1)							
H23(1)							

Note: The parenthesized numbers are sample quantities, and the underlined numbers indicate shared haplotypes between populations.

**Table 3 genes-14-00998-t003:** Genetic diversity indices of mtDNA control region within 8 Sichuan chicken breeds.

Breed	n	V	Ss	Ps	H	Hd	π	AND	Tajima’s D *
MH	199	28	3	25	23	0.876	0.012	5.936	0.567
JY	40	18	7	11	7	0.709	0.007	3.492	−0.571
MY	39	26	2	24	15	0.906	0.015	7.387	0.687
SM	46	19	0	19	7	0.741	0.010	4.817	0.367
MC	29	12	1	11	5	0.732	0.007	3.167	0.120
PX	27	20	8	12	7	0.641	0.011	5.054	0.289
JS	29	13	5	8	8	0.700	0.013	6.059	0.123
EM	30	17	8	9	10	0.786	0.010	4.784	0.393
Total	439	46	7	39	44	0.895	0.013	6.349	−0.331

Note: * All of them were not statistically significant in this study (*p*-values > 0.1), and some of the data in this table came from our previous study [30]. Variable sites: V; singleton sites: Ss, parsimony informative sites: Ps; haplotypes: H; haplotype diversity: Hd; nucleotide diversity: π; average no. of nucleotide differences: AND.

**Table 4 genes-14-00998-t004:** Analysis of molecular variance (AMOVA) of 439 mtDNA D-loop sequences from 8 local chicken breeds in Sichuan province.

Sources of Variation	d.f.	Sum of Squares	Variance Components	Percentage of Variation	*Fst*	*p*-Value
Among populations	7	221.676	0.61549 Va	18.50	-	-
Within populations	431	1168.787	2.71180 Vb	81.50	-	-
Total	438	1390.462	3.32729	-	0.18498	0.00

Note: d.f., degree of freedom; *Fst*, fixation index; Va, among-population variance; Vb, between-individual within-population variance.

**Table 5 genes-14-00998-t005:** Genetic differentiation index (Gst), gene flow (Nst), nucleotide divergence (Dxy), and net genetic distance (Da) between each pair of chicken breeds.

	MH	JY	MY	SM	MC	PX	JS	EM
Coefficient of differentiation and gene flow								
MH		18.537	15.626	18.345	37.888	13.415	11.728	12.640
JY	4.793		27.880	37.464	51.426	3.860	4.010	7.071
MY	2.711	6.564		8.399	18.811	18.033	17.822	14.482
SM	3.483	12.530	6.804		10.451	23.185	22.530	16.239
MC	3.642	8.952	5.732	5.601		36.304	35.440	28.150
PX	4.751	2.851	8.370	16.865	12.470		−2.396	0.693
JS	4.261	2.539	7.440	15.898	11.815	−1.036		2.065
EM	3.031	1.753	3.917	5.264	3.349	5.285	4.972	
Nucleotide divergence and net genetic distance								
MH		0.225	0.258	0.252	0.578	0.179	0.167	0.163
JY	1.212		0.438	0.521	0.735	0.036	0.042	0.066
MY	1.652	1.576		0.119	0.257	0.285	0.303	0.215
SM	1.377	1.391	1.396		0.097	0.313	0.332	0.195
MC	1.530	1.431	1.361	0.932		0.489	0.528	0.324
PX	1.328	0.930	1.587	1.346	1.349		−0.027	0.007
JS	1.422	1.041	1.710	1.469	1.494	1.135		0.024
EM	1.284	0.932	1.488	1.199	1.156	1.036	1.158	

Note: The upper half is made up of the Gst (lower diagonal) and the Nst (upper diagonal). The lower half is made up of the Dxy (below diagonal) and the Da (above diagonal). All the values were enlarged 100 times, with the maximum and minimum values underlined. Relevant data of JY, MY, SM, MC, PX, JS, and EM came from our previous study [30].

**Table 6 genes-14-00998-t006:** Composition of haplogroups in MH chickens.

Clade	A	B	C	E	G
	H1(2)	H2(34)	-	H7(22)	H21(2)
	H5(17)	H3(45)	-	H12(13)	H23(1)
	H6(4)	H4(1)	-	H13(1)	-
	H10(28)	H8(1)	-	H16(6)	-
	H11(2)	H9(1)	-	H17(2)	-
	H14(3)	H15(1)	-	H20(10)	-
	H18(1)	-	-	-	-
	H19(1)	-	-	-	-
	H22(1)	-	-	-	-
Total	59	83	0	54	3
Ratio(%)	29.65%	41.71%	0	27.14%	1.50%

## Data Availability

The mitochondrial DNA control region data were submitted to NCBI with the GenBank accession numbers OP455159–OP455357.

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
