# Peer review of "Studying Genetic Diversity and Relationships between Mountainous Meihua Chickens Using Mitochondrial DNA Control Region"

_genes, 2023, doi:10.3390/genes14050998_

Round 1
Reviewer 1 Report
In this paper, the authors analyse the genetic diversity of the Mountainous Meihua (MH) chicken breed. For this purpose, they use the sample of 199 chickens and the highly variable D-loop region of mitochondrial DNA. MH chicken breed was compared with seven other breeds that are kept in the region. They have performed various analyses using named genetic marker. Performed analyses are typical for population genetics studies and conducted in a correct manner. The results showed the presence of four clades in MH chicken with 23 haplotypes, and some of these haplotypes seem to be specific to the MH chicken breed. Presented results show that the MH chicken breed possesses some unique features and that it should be considered a breed worthy of protection through conservation programs. These are my comments for the authors.
Introduction
Although short it provides necessary information. Maybe the information about the number of different breeds available in the region, or even China, would be nice to have here in order to understand why the MH breed was compared with seven other breeds. What is the relationship between MH breed and other breeds? It would be nice to stress the importance of this breed from the perspective of numbers: how many farms breed MH, what is the number of the birds, what is the revenue that the breeding of this breed provides compared to other breeds… This will stress why it is important to conduct research on this chicken breed.
Line 25: insert binomial nomenclature for the chicken, Galus galus domesticus, it should be located behind the name of the analysed chicken breed
Line 32: reference number 4 is wrong here. The listed reference is: “Du, C.; Chen, X. Transcriptome Profiling of Oocytes at the Germinal Vesicle Stage from Women from Mongolia with Polycystic Ovary Syndrome. Int J Gen Med 2021, 14, 4469-4478, doi:10.2147/IJGM.S321853”. This reference has nothing to do with the Meihua chicken. Please check your library and out the correct reference.
Line 49: move “and” in “genetic diversity and population structure and uncover”
Materials and methods
Line 61: How did you collect the blood samples, ie. from where did you extract the blood? Leg or wing vein?
Line 62: “mountain plum blossom” Is this the name of the institution for chicken breeding?
Line 70: everywhere where you have “mL” you should correct to µl
Line 71: how many ng of DNA did you put in each reaction for amplification? Put that information after “genomic DNA”
Lines 80-83: Reference the papers from which these sequences originate
Line 90: what other indicators, please name them since you are using them later in the text, more precisely in the results section
Line 92: “Fst” should be in italics like this FST with upper cases in the index
Figure 1. Add a scale to the figure so that readers unfamiliar with the region can have a sense of the distances
Results
In the results section I would like to see the analysis of pairwise FST within the pairs of populations and these results can also be visualized using multidimensional scaling (MDS). Software Past 4.03 is a user-friendly program which you can use to visualize FST matrix with ease. Population pairwise FST values can be calculated using the Arlequin software. Maybe this can resolve the issue of the conflicting results of Dxy and Gst (more about this in the comments for the Discussion section).
Line 129: breeds instead of “species”
Table 3. correct all the names in the first row, the abbreviations should be listed and the legend should be below the table. Variable sites replace with V; Singleton sites for example replace with Ss; Parsim-info sites for example replace with Ps; Haplotypes replase with H; Haplotype diversity replace with Hd; Nucleotide diversity replace with π; Average number of nucleotide differences replace with K. You can also add parameter Mean Pairwise Differences (MPD) which can be calculated in the Arlequin software.
Line 138: is it population or breed? In this case, you can say that the sample of each breed represents a population from that specific breed, but since you have one population for each breed maybe you should use the term breed here. And change it throughout the text
Table 5. put the Dxy-Da matrix first and Gst-Nst matrix second, or reorder the text in the Table title to correspond to the order in the table
Figure 4. You can try the program Network and perform the median-joining analysis with maximum parsimony that will remove the reticulation in the network. With Network software, you can also identify the mutations as the nucleotide positions on which the substitution occurred. This will create a more readable network.
Discussion
Line 197: What does A/T bias tell us? Is it significant for this type of analysis and comparison of different breeds of the same species, since it is located in the D-loop which should be selectively neutral? My point is that it is a specific feature of the control region which applies to all species so why it is relevant to mention this information in light of these analyses? Should we expect it to be different in any way among different breeds of the same species?
Line 197: replace “species” with breeds
Line 198-199: This sentence needs to be clarified. What about the detected genetic variation? Is it a newly detected variation? Replace populations with breeds, see comment for line 138.
Line 205: “higher than most other analyzed breeds analyzed.” analyzed goes before breeds. Be careful with the interpretation here, you also had a greater number of birds analyzed for the MH breed compared to other breeds ie. you have unbalanced sample sizes. In order to properly compare the samples with different sizes you should use the rarefaction method.
Line 206-207: I would also be careful here, it is OK to compare your results with the results from the studies that used the same region that was amplified with the same primer pairs as you did, just like in Sha 2019., but the study of Jia 2017 used the complete D-loop sequences. It is better to use the actual sequences that are available through their accession numbers in your analyses and in that way you can properly compare them with your samples.
Lines 218-220: How would you explain this finding? It seems that the geographically close populations of these breeds are genetically most distant while geographically distant populations of analyzed breeds are genetically more similar. Does it have something to do with the way how these breeds are managed? Is the history of these breeds known? What was the original breed from which other breeds were selected and bred, and when was each of the breeds established as a distinctive breed? Do the breeders exchange birds from different breeds for selection purposes?
Line 222: “Our results showed high differences between MH and MY,” this directly contradicts the finding that the MH and MY are genetically most similar ie. with the lowest level of genetic differentiation (line 219, table 5). These two parameters should be concordant and not contradictory. Check your results again, maybe you mixed the numbers in table 5.
Line 235-236: “could result from admixture with local Yunnan varieties”. Interestingly, this clade is represented by two haplotypes. One is the most frequent clade G haplotype and the other (detected only in MH) diverges from it and can be identified on the basis of one substitution. This could be indicative that this admixture happened a long time ago since the new haplotype detected only in MH had time to evolve.
Line 241: “three maternal lineages”, what about G? It is also present in the MH breed.
Line 242, 244: replace “species” with breeds
Some of the sentences are not clear and in some places word order should be replaced. I recommend that someone with good English skills check the text prior to the submission.
Author Response
Dear Editor Josie Long,
Thank you very much for processing our manuscript “Genetic diversity and relationships of Mountainous Meihua chickens using mitochondrial DNA control region” (ID: genes-2344810) for possible publication in Genes. We are very grateful to the editor and two reviewers for their helpful suggestions and positive comments and for giving us an opportunity to revise the manuscript. We have carefully considered the comments and have made corrections that we hope will meet with the criterion of Genes. For all modifications, please refer to the section highlighted in red in the revised manuscript. Our point-by-point responses to the reviewer's comments are presented following.
Response to comments of the editor:
- Please add fully experimental details and present completely all the results.
Response: As suggested, we have added fully experimental details in our revised manuscript (Line 90-98).
- In the introduction section, provide comprehensively the background and overview of the research. Please ensure that the text of the revision exceeds 4000 words and the page number exceeds 10 pages.
Response: We sincerely appreciate the valuable comments. In the introduction section (Line 36-70), we have added more information to describe in detail the application of mitochondrial D-loop studies in poultry. about mitochondrial research. Secondly, we further enrich the discussion of our findings. We also discussed more in our revised manuscript (Lines 92-100, 243-261 etc.). All of these changes ensured that the revision exceeds 4000 words and the page number exceeds 10 pages.
Line 36-70: As an important extracellular genetic information carrier, mitochondrial DNA (mtDNA) has low relative molecular weights, high copy numbers, maternal inheritance, and no/rare recombination events, which was wildly used as a marker in many molecular studies[1-4]. It is a popular choice for phylogenetic, population genetic, and phylogeographic analyses[5]. The mtDNA D-loop region is a non coding region, rich in A/T bases, which can be recognized by specific molecules such as RNA polymerase and used as the starting sequence of transcription[6]. In the mitochondrial genome, coding genes are mainly used for biodiversity research of different species, while non coding D-loop regions are commonly used for genetic diversity analysis and maternal origin research due to their evolution speed being 5-10 times higher than other regions on mtDNA[7-9].
Using mtDNA information, previous studies indicated that domestic chickens singlely originated from a group of red jungle poultry found in forests in Southeast Asia and India, and gradually distributed throughout the world with continuous domestication and transmission, resulting in many new breeds[10]. Research on the origin of Chinese domestic chicken breeds revealed that the red jungle fowl from Thailand and its neighboring areas are considered as the maternal ancestors of Chinese chicken breeds[11]. However, a series of independent domestication events across Asia and the Indian subcontinent, supporting multiple origins of domestic chickens[12,13]. In 2006, Liu et al[14] first proposed that domestic chickens in Eurasia are distributed in nine highly different mtDNA evolutionary branches (A-I) and pointed out different regions where different branches may originate. Later, another research proposed more branches and further determined the maternal origin of domestic chickens[15]. The analysis of the origin of Japanese indigenous chickens revealed that the wild red jungle fowl was domesticated in Southeast Asia and passed into China to form non wild clades (A and B), which together form the foundation of Japanese indigenous chickens[16]. Research on the origins and transmission routes of domestic chickens in the Middle East and Africa supports the hypothesis that the Indian subcontinent has a significant effect on the maternal origin of these chickens. Additionally, ancient and modern trade networks contributed to the modern diversity of native chickens[17]. A growing number of studies on mtDNA D-loop regions have revealed genetic diversity within populations and inter population relationships of some new local chicken breeds in various regions in recent years[18-21], which providing important support for modern breeding or chicken genetic diversity protection plans.
Detailed response to reviewers:
Reviewer #1
In this paper, the authors analyse the genetic diversity of the Mountainous Meihua (MH) chicken breed. For this purpose, they use the sample of 199 chickens and the highly variable D-loop region of mitochondrial DNA. MH chicken breed was compared with seven other breeds that are kept in the region. They have performed various analyses using named genetic marker. Performed analyses are typical for population genetics studies and conducted in a correct manner. The results showed the presence of four clades in MH chicken with 23 haplotypes, and some of these haplotypes seem to be specific to the MH chicken breed. Presented results show that the MH chicken breed possesses some unique features and that it should be considered a breed worthy of protection through conservation programs. These are my comments for the authors.
Response: Thanks for the reviewer’s positive comments.
Introduction
Although short it provides necessary information. Maybe the information about the number of different breeds available in the region, or even China, would be nice to have here in order to understand why the MH breed was compared with seven other breeds. What is the relationship between MH breed and other breeds? It would be nice to stress the importance of this breed from the perspective of numbers: how many farms breed MH, what is the number of the birds, what is the revenue that the breeding of this breed provides compared to other breeds… This will stress why it is important to conduct research on this chicken breed.
Response: There is 1 protected population, 68 protected families and more than 2000 pure breeds of MH chicken in Bazhong Mountain, so wo collected the samples from this protected population. There are 7 chicken breeds in book “Local livestock and poultry breeds in Sichuan”. Thus we compared these 7chicken breeds with our MH chickens.
Line 25: insert binomial nomenclature for the chicken, Galus galus domesticus, it should be located behind the name of the analysed chicken breed
Response: Corrected as suggested.
Line 32: reference number 4 is wrong here. The listed reference is: “Du, C.; Chen, X. Transcriptome Profiling of Oocytes at the Germinal Vesicle Stage from Women from Mongolia with Polycystic Ovary Syndrome. Int J Gen Med 2021, 14, 4469-4478, doi:10.2147/IJGM.S321853”. This reference has nothing to do with the Meihua chicken. Please check your library and out the correct reference.
Response: Thanks for the comment. We have replaced the reference as “Du H, Sun D-F, Yi X, Xu D, Liao J-H, Wang F, Shu G. 2021. Effects of Probiotics Combined with Fermentation Bed Culture Model on Growth Performance, Slaughter Performance, Immunologic Function and Intestinal Flora of Bazhong Mountain Plum Chicken. Feed Industry. 42(12):6-10.”
Line 49: move “and” in “genetic diversity and population structure and uncover”
Response: Changed as suggested.
Materials and methods
Line 61: How did you collect the blood samples, ie. from where did you extract the blood? Leg or wing vein?
Response: Thanks for the comment. Blood samples were collected from 199 male adult Mountainous Meihua chickens via wing vein.
Line 62: “mountain plum blossom” Is this the name of the institution for chicken breeding?
Response: Yes, it is the name of the institution for chicken breeding, we have revised that in our new manuscript.
Line 70: everywhere where you have “mL” you should correct to µl
Response: Thanks for the comment. Changed as suggested.
Line 71: how many ng of DNA did you put in each reaction for amplification? Put that information after “genomic DNA”
Response: Added as suggested.
Lines 80-83: Reference the papers from which these sequences originate
Response: The reference was added as suggested.
Line 90: what other indicators, please name them since you are using them later in the text, more precisely in the results section
Response: Thanks for the comment. Changed as suggested. The indicators are Nucleotide divergence (Dxy), net genetic distance (Da), Genetic differentiation index (Gst) and gene flow (Nst), Haplotypes, and Average No. of nucleotide differences.
Line 92: “Fst” should be in italics like this FST with upper cases in the index
Response: Thanks for the comment. Changed as suggested.
Figure 1. Add a scale to the figure so that readers unfamiliar with the region can have a sense of the distances
Response: Thanks for the comment. Changed as suggested.
Results
In the results section I would like to see the analysis of pairwise FST within the pairs of populations and these results can also be visualized using multidimensional scaling (MDS). Software Past 4.03 is a user-friendly program which you can use to visualize FST matrix with ease. Population pairwise FST values can be calculated using the Arlequin software. Maybe this can resolve the issue of the conflicting results of Dxy and Gst (more about this in the comments for the Discussion section).
Response: Thanks for the helpful suggestion, we will try this recommended software in further studies. Because time limited, we are sorry for not reanalyzing our data using Software Past 4.03.
Line 129: breeds instead of “species”
Response: Thanks for the comment. Changed as suggested.
Table 3. correct all the names in the first row, the abbreviations should be listed and the legend should be below the table. Variable sites replace with V; Singleton sites for example replace with Ss; Parsim-info sites for example replace with Ps; Haplotypes replase with H; Haplotype diversity replace with Hd; Nucleotide diversity replace with π; Average number of nucleotide differences replace with K. You can also add parameter Mean Pairwise Differences (MPD) which can be calculated in the Arlequin software.
Response: Thanks for the comment. Changed as suggested.
Line 138: is it population or breed? In this case, you can say that the sample of each breed represents a population from that specific breed, but since you have one population for each breed maybe you should use the term breed here. And change it throughout the text
Response: Thanks for the comment. It is breed. Yes, we have one population for each breed. We used the term breed in our revised manuscript as suggested.
Table 5. put the Dxy-Da matrix first and Gst-Nst matrix second, or reorder the text in the Table title to correspond to the order in the table
Response: Thanks for the comment. We have reordered the text in the Table title as suggested.
Figure 4. You can try the program Network and perform the median-joining analysis with maximum parsimony that will remove the reticulation in the network. With Network software, you can also identify the mutations as the nucleotide positions on which the substitution occurred. This will create a more readable network.
Response: Thanks for the helpful suggestion, we will try this recommended software in further studies. Because time limited, we are sorry for not reanalyzing our data using the program Network.
Discussion
Line 197: What does A/T bias tell us? Is it significant for this type of analysis and comparison of different breeds of the same species, since it is located in the D-loop which should be selectively neutral? My point is that it is a specific feature of the control region which applies to all species so why it is relevant to mention this information in light of these analyses? Should we expect it to be different in any way among different breeds of the same species?
Response: Thanks for the comment. Base bias is a common feature of mitochondrial D-loop sequences in all chicken breeds. But here, we focus on considering that it is necessary to point out the A/T bias when first describing the mitochondrial D-loop region characteristics of these 8 chicken breeds.
Line 197: replace “species” with breeds
Response: Thanks for the comment. Changed as suggested.
Line 198-199: This sentence needs to be clarified. What about the detected genetic variation? Is it a newly detected variation? Replace populations with breeds, see comment for line 138.
Response: Thanks for the comment. We have modified this part of the content. Yes, the genetic variation we detected belongs to new genetic variation. “Populations” has been replaced by “breeds”.
Line 205: “higher than most other analyzed breeds analyzed.” analyzed goes before breeds. Be careful with the interpretation here, you also had a greater number of birds analyzed for the MH breed compared to other breeds ie. you have unbalanced sample sizes. In order to properly compare the samples with different sizes you should use the rarefaction method.
Response: Thanks for the comment. Yes, because the number size of different breeds is different, we have modified related description.
Line 206-207: I would also be careful here, it is OK to compare your results with the results from the studies that used the same region that was amplified with the same primer pairs as you did, just like in Sha 2019., but the study of Jia 2017 used the complete D-loop sequences. It is better to use the actual sequences that are available through their accession numbers in your analyses and in that way you can properly compare them with your samples.
Response: Thanks for the comment. We have removed references that used different primers for the amplification area.
Lines 218-220: How would you explain this finding? It seems that the geographically close populations of these breeds are genetically most distant while geographically distant populations of analyzed breeds are genetically more similar. Does it have something to do with the way how these breeds are managed? Is the history of these breeds known? What was the original breed from which other breeds were selected and bred, and when was each of the breeds established as a distinctive breed? Do the breeders exchange birds from different breeds for selection purposes?
Response: Thanks for the comment. It is not strange that the geographically close populations of these breeds are genetically most distant, because their gene exchange might be prevented by higher mountains or rivers. And most of these breeds are free-ranging in China. The history of most of these breeds are unknown. Because of breeders exchange birds from different breeds for selection purposes, there are gene flow among these breeds.
Line 222: “Our results showed high differences between MH and MY,” this directly contradicts the finding that the MH and MY are genetically most similar ie. with the lowest level of genetic differentiation (line 219, table 5). These two parameters should be concordant and not contradictory. Check your results again, maybe you mixed the numbers in table 5.
Response: Thanks for the comment. Dxy refers to the ability to have several different nucleotide choices for a particular location in a DNA or RNA sequence. For example, for A single base location, if four different nucleotides A, T, C, and G may be present, the nucleotide ambiguity at that location is 4. This does not completely correlate with Gst. We have modified related description, and only retained Gst discussion.
Line 235-236: “could result from admixture with local Yunnan varieties”. Interestingly, this clade is represented by two haplotypes. One is the most frequent clade G haplotype and the other (detected only in MH) diverges from it and can be identified on the basis of one substitution. This could be indicative that this admixture happened a long time ago since the new haplotype detected only in MH had time to evolve.
Response: Thanks for the helpful and positive comment.
Line 241: “three maternal lineages”, what about G? It is also present in the MH breed.
Response: Thanks for the comment. Changed as suggested.
Line 242, 244: replace “species” with breeds
Response: Thanks for the comment. Changed as suggested.
Reviewer 2
The authors of study entitled “Genetic diversity and relationships of Mountainous Meihua chickens using mitochondrial DNA control region” aimed to evaluate the genetic diversity and population structure of 439 mtDNA control-region sequences belonging to Mountainous Meihua chickens and other seven local breeds. The paper is quite well written in general. However, there are some points that must be better explained and deepened.
Major comments:
Abstract:
I am not sure to have really caught how many samples were analyzed from authors and how many sequences were retrieved from GenBank. In the abstract, authors stand that 439 mtDNA control-region sequences were generated: “199 from Mountainous Meihua chicken and 240 from seven distinct breeds”, but in the M&M section they describe the Ethical statement, the Sample collection and DNA extraction only for Mountainous Meihua chickens. So, if authors did not generate all 439 sequences, I think they should modify the abstract.
Response: Thanks for the comment. We have replaced the part “Here, we generated a total of 439 mitochondrial sequences (199 from Mountainous Meihua chicken and 240 from seven distinct breeds) and downloaded 30 mtDNA D-loop regions” as “Here, we analyzed a total of 469 sequences, including 199 Mountainous Meihua chicken sequences generated in this study, 30 sequences representing 13 clades and 240 sequences from 7 different Sichuan local chicken breeds downloaded from NCBI” to explain sample source.
Materials and Methods:
In a phylogenetic/phylogeographic study, the sample collection is a key point, thus I think authors should describe the selection criteria used to collect samples. Have you considered the maternal relationships among individuals, or they were impossible to be evaluated? I think you should specify also if you collected samples from all chickens, or sampled only a part of the breeding base, or if it was a random sampling.
Response: Thanks for the comment. We have corrected the description of the sample collection as line 92-94.
Moreover, do you think that 199 samples can be better compared with other breeds each represented by 30-40 individuals? How did you choose these published sequences and the other 30 mtDNA control regions? Please explain.
Response: Thanks for the comment. We selected as many samples as possible from Mountainous plum chickens to more accurately characterize their genetic diversity. The mitochondrial control region sequences of the seven local Sichuan chicken breeds were selected with reference to Li's research[22]. Another 30 mitochondrial control region sequences were randomly selected from the sequences used in the study of seedlings, representing 13 branches[15].
Minor comments:
Throughout the text: numbers from one to ten are usually written in letters
Line 12: remain
Response: Thanks for the comment. Changed as suggested.
Line 13: please, specify that you have analyzed 439 mtDNA control-region sequences
Response: Thanks for the comment. In fact, we analyzed a total of 469 sequences. Based on your suggestions, we specify the 469 sequences we analyzed and explain the source of these sequences.
Line 14: replace “downloaded” with “compared them with”
Response: Thanks for the comment. We have corrected the description of this section and we think it is appropriate to use 'download' in the corrected content.
Line 18: chickens belong to
Response: Thanks for the comment. Changed as suggested.
Lines 27: showed that
Response: Thanks for the comment. Changed as suggested.
Line 34: replace “that allow for” with “allowing for”
Response: Thanks for the comment. Changed as suggested.
Line 36: replace “study” with “analyze”
Response: Thanks for the comment. We have changed the content in this part, and the changed content does not involve the use of "study".
Line 39: replace “which has also been shown in chickens” with “as also shown for chickens”
Response: Thanks for the comment. We have changed the content in this part, and the changed content does not involve the use of “which has also been shown in chicken”.
Line 63: “All samples were stored at -20°C to preserve mitochondrial DNA”: this is to preserve all the genetic material (genomic DNA)
Response: Thanks for the comment. Changed as suggested.
Line 74: What did authors use to purify PCR products? And what primers were used for the sequencing?
Response: Thanks for the comment. The PCR products were purified using the gel recovery kit. Sequencing primers and PCR amplification primers were consistent. The forward primer (5’-AGGACTACGGCTTGAAAAGC-3’) and reverse primer (5’-ATGTGCCTGACCGAGGAACCAG
-3’).
Line 117-118: There is something wrong in the sentence “The haplotypes found in MH mainly belonged to the H2, H3, H5, H7, H10, H12 and H20 haplogroups”. These are not HAPLOGROUPS, but HAPLOTYPES. Please, re-write the sentence
Response: Thanks for the comment. We have corrected the description as “The haplotypes found in MH mainly include the H2, H3, H5, H7, H10, H12 and H20”.
Line 128 and 135: It could be better referring to this table with “Genetic diversity indices” or something similar, instead of “nucleotide polymorphism”
Response: Thanks for the comment. Changed as suggested.
Line 165: “MEGA5.0 software”. In M&M section (Line 93) authors declared that they used MEGA 6.0 software. Please correct the mistake.
Response: Thanks for the comment. Changed as suggested.
Line 176: GenBank
Response: Thanks for the comment. Changed as suggested.
Line 180: replace “Conposition” with “Haplotype composition”
Response: Thanks for the comment. Changed as suggested.
Line 190: 482 bp control-region sequences
Response: Thanks for the comment. Changed as suggested.
Line 203: Haplotype and nucleotide diversity
Response: Thanks for the comment. Changed as suggested.
Lines 203-204: too many repetitions of genetic diversity. Please use variation, variability, ………..
Response: Thanks for the comment. Changed as suggested.
Line 204: delete “a” after “We found”
Response: Thanks for the comment. Changed as suggested.
Line 205: Move “in MH” at the beginning of the sentence
Response: Thanks for the comment. Changed as suggested.
Line 206: than most of local breeds here analyzed
Response: Thanks for the comment. Changed as suggested.
Line 208: please, briefly explain in the text why your results could provide theoretical support for improving breeding practices, as the sentence can be clear also to not expert readers
Response: Thanks for the comment. We have added some content earlier in this paragraph to illustrate the significance of genetic diversity in breeding.
Line 213: remove the space before the dot
Response: Thanks for the comment. Changed as suggested.
Line 226: I think there are also more recent references that should be cited here regarding the geographic isolation and breeding histories of different species (i.e.: Toalombo et al. 2019. Deciphering the Patterns of Genetic Admixture and Diversity in the Ecuadorian Creole Chicken; Giontella et al. 2020. s; Xia et al .2020. Mitogenome Diversity and Maternal Origins of Guangxi Cattle Breeds; Lancioni et a. 2016. Survey of uniparental genetic markers in the Maltese cattle breed reveals a significant founder effect but does not indicate local domestication; Kusliy et al. 2021. Traces of Late Bronze and Early Iron Age Mongolian Horse Mitochondrial Lineages in Modern Populations)
Response: Thanks for the comment. Changed as suggested. According to your suggestion, we have added these references in the paper.
Line 227: According to
Response: Thanks for the comment. Changed as suggested.
Line 227: I guess authors meant that “the mtDNA phylogenetic tree of domestic chickens can be divided into 13 clades”. Please, correct the text
Response: Thanks for the comment. As described in the original reference “Chicken domestication: an updated perspective based on mitochondrial genomes” for 13 clades, common haplogroups A–G were shared by domestic chickens and red junglefowl. Rare haplogroups H–I and W–Z were specific to domestic chickens and red junglefowl, respectively. So we think the description of this part is correct.
Lines 228-231: please, provide some references
Response: Thanks for the comment. Like the source of the first sentence above, this part of content is also from the literature of "Liu [19] Miao [20]".
Line 234: I can see another font size for this line; is this true? If yes, please correct
Response: Thanks for the comment. We carefully checked the font size for this line and found no other font size.
Line 237: belong to haplogroup B
Response: Thanks for the comment. Changed as suggested.
Line 239: has a relatively
Response: Thanks for the comment. Changed as suggested.
Lines 242 and 244: replace “species” with “breed” or “local breed”
Response: Thanks for the comment. Changed as suggested.
Reviewer 2 Report
The authors of study entitled “Genetic diversity and relationships of Mountainous Meihua chickens using mitochondrial DNA control region” aimed to evaluate the genetic diversity and population structure of 439 mtDNA control-region sequences belonging to Mountainous Meihua chickens and other seven local breeds. The paper is quite well written in general. However, there are some points that must be better explained and deepened.
Major comments:
Abstract:
I am not sure to have really caught how many samples were analyzed from authors and how many sequences were retrieved from GenBank. In the abstract, authors stand that 439 mtDNA control-region sequences were generated: “199 from Mountainous Meihua chicken and 240 from seven distinct breeds”, but in the M&M section they describe the Ethical statement, the Sample collection and DNA extraction only for Mountainous Meihua chickens. So, if authors did not generate all 439 sequences, I think they should modify the abstract.
Materials and Methods:
In a phylogenetic/phylogeographic study, the sample collection is a key point, thus I think authors should describe the selection criteria used to collect samples. Have you considered the maternal relationships among individuals, or they were impossible to be evaluated? I think you should specify also if you collected samples from all chickens, or sampled only a part of the breeding base, or if it was a random sampling.
Moreover, do you think that 199 samples can be better compared with other breeds each represented by 30-40 individuals? How did you choose these published sequences and the other 30 mtDNA control regions? Please explain.
Minor comments:
Throughout the text: numbers from one to ten are usually written in letters
Line 12: remain
Line 13: please, specify that you have analyzed 439 mtDNA control-region sequences
Line 14: replace “downloaded” with “compared them with”
Line 18: chickens belong to
Lines 27: showed that
Line 34: replace “that allow for” with “allowing for”
Line 36: replace “study” with “analyze”
Line 39: replace “which has also been shown in chickens” with “as also shown for chickens”
Line 63: “All samples were stored at -20°C to preserve mitochondrial DNA”: this is to preserve all the genetic material (genomic DNA)
Line 74: What did authors use to purify PCR products? And what primers were used for the sequencing?
Line 117-118: There is something wrong in the sentence “The haplotypes found in MH mainly belonged to the H2, H3, H5, H7, H10, H12 and H20 haplogroups”. These are not HAPLOGROUPS, but HAPLOTYPES. Please, re-write the sentence
Line 128 and 135: It could be better referring to this table with “Genetic diversity indices” or something similar, instead of “nucleotide polymorphism”
Line 165: “MEGA5.0 software”. In M&M section (Line 93) authors declared that they used MEGA 6.0 software. Please correct the mistake.
Line 176: GenBank
Line 180: replace “Conposition” with “Haplotype composition”
Line 190: 482 bp control-region sequences
Line 203: Haplotype and nucleotide diversity
Lines 203-204: too many repetitions of genetic diversity. Please use variation, variability, ………..
Line 204: delete “a” after “We found”
Line 205: Move “in MH” at the beginning of the sentence
Line 206: than most of local breeds here analyzed
Line 208: please, briefly explain in the text why your results could provide theoretical support for improving breeding practices, as the sentence can be clear also to not expert readers
Line 213: remove the space before the dot
Line 226: I think there are also more recent references that should be cited here regarding the geographic isolation and breeding histories of different species (i.e.: Toalombo et al. 2019. Deciphering the Patterns of Genetic Admixture and Diversity in the Ecuadorian Creole Chicken; Giontella et al. 2020. A Genetic Window on Sardinian Native Horse Breeds through Uniparental Molecular Systems; Xia et al .2020. Mitogenome Diversity and Maternal Origins of Guangxi Cattle Breeds; Lancioni et a. 2016. Survey of uniparental genetic markers in the Maltese cattle breed reveals a significant founder effect but does not indicate local domestication; Kusliy et al. 2021. Traces of Late Bronze and Early Iron Age Mongolian Horse Mitochondrial Lineages in Modern Populations)
Line 227: According to
Line 227: I guess authors meant that “the mtDNA phylogenetic tree of domestic chickens can be divided into 13 clades”. Please, correct the text
Lines 228-231: please, provide some references
Line 234: I can see another font size for this line; is this true? If yes, please correct
Line 237: belong to haplogroup B
Line 239: has a relatively
Lines 242 and 244: replace “species” with “breed” or “local breed”
Author Response

(The authors gave the same response as above.)
